# Building indoor model in PALM-4U: Indoor climate, energy demand, and the interaction between buildings and the urban microclimate

Jens Pfafferott[1], Sascha Rißmann[1], Matthias Sühring[2], Farah Kanani-Sühring[3], and Björn Maronga[2]

[1]Offenburg University of Applied Sciences, Institute of Energy Systems Technology, Offenburg, 77652, Germany
[2]Leibniz University Hannover, Institute of Meteorology and Climatology, Hannover, 30419, Germany
[3]Harz Energie GmbH & Co. KG, Goslar, 38640, Germany

*Correspondence to*: Jens Pfafferott (jens.pfafferott@hs-offenburg.de)

**Abstract.** There is a strong interaction between the urban atmospheric canopy layer and the building energy balance. The
urban atmospheric conditions affect the heat transfer through exterior walls, the longwave heat transfer between the building surfaces and the surroundings, the shortwave solar heat gains and the heat transport by ventilation. Considering also the internal heat gains and the heat capacity of the building structure, the energy demand for heating and cooling and the indoor thermal environment can be calculated based on the urban microclimatic conditions. According to the building energy concept, the energy demand results in an (anthropogenic) waste heat, this is directly transferred to the urban environment.
Furthermore, the indoor temperature is re-coupled via the building envelope to the urban environment and affects indirectly the urban microclimate with a temporally lagged and damped temperature fluctuation. We developed a holistic building model for the combined calculation of indoor climate and energy demand based on an analytic solution of Fourier's equation and implemented this model into the PALM model.

## 1 Introduction: Interaction between the building and the urban atmospheric conditions in simulation

Buildings strongly affect the urban atmospheric conditions. Reversely, the urban microclimate strongly affects the indoor climate and energy demand of buildings. A good review on experimental and numerical studies from the 1960s to today is given by Helbig et al. (2013). Hence, urban microclimate simulation models should contain a powerful building indoor model in order to evaluate the strong interaction between the building and the urban atmospheric conditions.

In a preliminary simulation study, Jacob and Pfafferott (2012) applied different test reference years (Deutscher Wetterdienst,
2014) on different building concepts and operation strategies. These test reference years consider both the climate change and the urban microclimate effect. The study clearly revealed that the urban heat island effect has a stronger effect on the building energy balance than the climate change. As expected, the building physical parameters of the building envelope (i.e. heat transfer coefficients, window area related to façade and floor area, fabric, solar shading) and the user behaviour (i.e. attendance, ventilation, use of shading device) strongly affects energy demand in summer and winter and indoor

environment for both residential and office buildings. Results from monitoring campaigns confirm these findings, (Kalz et al., 2014) and (Pfafferott and Becker, 2008).

Favourably, those complex interactions between the built environment and the urban atmospheric conditions can be evaluated based on a sophisticated simulation model (Bueno et al., 2012). Within the MOSAIK project (Maronga et al., 2019), we developed a holistic building model for the coupled calculation of indoor climate (i.e. operative room temperature)

and energy demand for heating, cooling, lighting and ventilation. The building indoor model is based on an analytic solution of Fourier's equation and is directly integrated into the PALM code via an interface with the building surface model (Maronga, et al., 2015, and Maronga, et al., 2020):

- The temperature of the building's boundary layer is calculated by PALM by extrapolation of the logarithmic law-of-the-wall temperature profile between the surface and the first node in the atmosphere depending on grid spacing employed

in the atmospheric component and the buoyancy effect at vertical and horizontal structures (Resler et al., 2017, and Maronga, et al., 2020). This temperature is the input variable for the calculation of heat transport by (free or mechanical) ventilation.

- The building indoor model gets the locally allocated wall, window and roof temperature of the innermost layer of each surface as an input from the building surface model for the simulation of indoor climate and energy demand.

- The building surface model gets the specific heat flux through the exterior walls as an input from the building indoor model for the simulation of façade temperatures.

The building surface model is based on a R5C4 resistance-capacity model in order to represent typical building structures with up to four wall layers. This model can be run within the PALM model with or without the building indoor model. If the building surface model is run in combination with the building indoor model, the indoor surface temperature is substituted

by the heat flux into the wall. In this case, the building surface model is reduced consequently to a R4C4 model.

**2 Development of a building indoor model for urban microclimate simulations**

Fourier's law in inhomogeneous materials and with heat sources can be shown in Cartesian coordinates x, y and z using the Nabla operator:

$$\rho(x,y,z) \cdot c(x,y,z) \cdot \frac{\partial T(x,y,z,t)}{\partial t} = \nabla \left[ \lambda(x,y,z) \cdot \nabla T(x,y,z,t) \right] + q(x,y,z,t) \tag{1}$$

with the density $\rho$, the specific heat capacity c, the heat conductivity $\lambda$ and the heat source $q$. If the different building materials are considered as one mean value for all components, the properties are homogeneous. Furthermore, constant heat transfer coefficients are a feasible assumption in buildings. Starting from these assumptions and considering the heat sources and the heat transfer, respectively, the partial differential equation of the thermal conduction equation can be solved analytically.

Thus, the building indoor model is based on an analytical solution of Fourier's law considering a resistance model with five resistances R [K W$^{-1}$] and one heat capacity C [J K$^{-1}$]. The solution is based on a Crank-Nicolson scheme for a one-hour time

step. As the programming is based on heat transfer coefficients H [W K$^{-1}$], all equations are based on the reciprocal value of R [K W$^{-1}$]. The heat transfer coefficient H takes short wave, long wave, convective and conductive heat transfer and heat transport (by air) into account.

The model considers four driving heat fluxes:

–    $\Phi_{hc}$ heating and cooling energy,

–    $\Phi_{conv}$ convective internal heat gains,

–    $\Phi_{rad,s}$ radiative internal and solar heat gains to the room-enclosing surfaces, and

–    $\Phi_{rad,m}$ radiative internal and solar heat gains to the room-enclosing building structure.

The solar radiation on each façade is calculated by the radiation model (Krc et al., 2020) and is passed through the building surface model to the building indoor model. All heat sources and sinks are coupled with

–    $\vartheta_i$ indoor air temperature,

–    $\vartheta_s$ interior surface temperature, or

–    $\vartheta_m$ temperature of the room-enclosing building structure, respectively.

These interior temperatures are coupled with three exterior temperatures:

–    $\vartheta_n$ temperature of the building's boundary layer (from the PALM model) for the incoming air,

–    $\vartheta_e$ ambient air temperature (from the PALM model) for the calculation of the heat transfer through the window, and

–    $\vartheta_w$ wall temperature (from the building surface model).

Figure 1 shows all temperatures [K] and driving heat flows [W] in this R5C1 network. From a numerical perspective, this

network consists of five reciprocal heat transfer coefficients H and one heat storage capacity C:

–    $H_v$ [W K$^{-1}$] for heat transport by ventilation between surface-near exterior air $\vartheta_n$ and indoor air $\vartheta_i$

–    $H_{t,is}$ [W K$^{-1}$] for convective heat transfer between indoor air $\vartheta_i$ and interior surfaces $\vartheta_s$ with specific heat transfer coefficient $h_{is}$=3.45 W m$^{-2}$ K$^{-1}$ considering all room-enclosing surfaces

–    $H_{t,es}$ [W K$^{-1}$] for heat transfer through windows between exterior air $\vartheta_e$ and interior surfaces $\vartheta_s$

–    $H_{t,ms}$ [W K$^{-1}$] for conductive heat transfer between interior surfaces $\vartheta_s$ and interior mass node $\vartheta_m$ with specific heat transfer coefficient $h_{ms}$=9.1 W m$^{-2}$ K$^{-1}$) considering room-enclosing surfaces

–    $H_{t,wm}$ [W K$^{-1}$] for conductive heat transfer between wall $\vartheta_w$ and interior mass node $\vartheta_m$

–    C [J K$^{-1}$] heat-storage capacity of all the whole room-enclosing building structure

From an energy balance perspective, the heat gains consist of internal and solar heat gains and the indoor environment is

coupled with the exterior environment both by heat transport due to ventilation and heat transfer through the envelope. Ventilation and heat transfer might be either heat gains (e.g. at high ambient temperatures or high solar heat gains at the opaque façades) or heat losses (esp. at higher indoor than outdoor temperatures). Furthermore, the thermal inertia of the building structure results in a time shift between the actual heat balance and the indoor temperature.

Since the heat transfer through the façade is calculated by the building surface model, the heat transfer through the façade $\Phi_w$ is the interface between both models. The building indoor model differs between the heat transfer through (transparent) windows and (opaque) walls. The corresponding heat transfer coefficients are calculated from input parameters given by the model database, cf. Chapter 3:

– The heat transfer coefficient $H_{t,es}$ through transparent, light-weight elements is reversely calculated from $H_{t,ei}$ and the heat transfer coefficient $H_{t,is}$ between the indoor air and the room-enclosing building structure into account. $H_{t,ei}$ is the product of the overall heat transfer coefficient $U_{window}$ [W m$^{-2}$ K$^{-1}$] and the window area $A_{window}$ [m²].

– The heat transfer coefficient $H_{t,wm}$ through opaque elements with thermal inertia is reversely calculated from $H_{t,ms}$ and the and the heat transfer coefficient $H_{t,ws}$ between the wall and the room-enclosing building structure. $H_{t,wm}$ is calculated from the heat conductivity and the thickness of the inside wall layer and the wall area $A_{wall}$ [m²].

Fourier's equation is mathematically solved for a time-step of 1 hour. The temperature of the room-enclosing building structure $\vartheta_m$ is calculated from its value at the previous time step $\vartheta_{m,prev}$ and the overall heat flux into the room-enclosing building structure $\Phi_{m,tot}$ which is calculated from $\Phi_{hc}$, $\Phi_{conv}$, $\Phi_{rad,s}$, and $\Phi_{rad,m}$ according to ISO 13790 (2008).

$$\vartheta_m = \frac{\vartheta_{m,prev}\cdot\left(\frac{C}{3600\,s}-0.5\cdot(H_{t,m}+H_{t,wm})\right)+\Phi_{m,tot}}{\frac{C}{3600\,s}+0.5\cdot(H_{t,m}+H_{t,wm})} \tag{2}$$

with $H_{t,m} = \dfrac{1}{\frac{1}{H_{t,s}+H_{t,es}}+\frac{1}{H_{t,ms}}}$ and $H_{t,s} = \dfrac{1}{\frac{1}{H_v}+\frac{1}{H_{t,is}}}$

The surface temperature $\vartheta_s$ is a function of the convective and radiative heat fluxes to the surface ($\Phi_{hc}$, $\Phi_{conv}$, and $\Phi_{rad,s}$) and is connected with the temperature of the room-enclosing building structure $\vartheta_m$, the temperature of the building's boundary layer $\vartheta_n$, and the ambient air temperature $\vartheta_e$.

$$\vartheta_s = \frac{H_{t,ms}\cdot\vartheta_m+\Phi_{rad,s}+H_{t,es}\cdot\vartheta_e+H_{t,s}\cdot\left(\vartheta_n+\frac{\Phi_{conv}+\Phi_{hc}}{H_v}\right)}{H_{t,ms}+H_{t,es}+H_{t,s}} \tag{3}$$

The indoor air temperature $\vartheta_i$ is a function of convective heat fluxes $\Phi_{hc}$ and $\Phi_{conv}$ and is coupled to the surface temperature $\vartheta_s$ and the $\vartheta_n$ near-façade temperature.

$$\vartheta_i = \frac{H_{t,is}\cdot\vartheta_s+H_v\cdot\vartheta_n+\Phi_{conv}+\Phi_{hc}}{H_{t,is}+H_v} \tag{4}$$

From these equations, the specific heating / cooling energy demand $\varphi_{hc,nd}$ [W m$^{-2}$] can be calculated for a specified set temperature for the indoor air $\vartheta_{i,set}$. This calculation is based on a linear approach based on the indoor air temperature without heating / cooling $\vartheta_{i,0}$ and the indoor air temperature $\vartheta_{i,10}$ with a specific heat flux $\varphi_{hc,10}$ of 10 W m$^{-2}$ net floor area $A_{nfa}$ [m²].

$$\varphi_{hc,nd} = \varphi_{hc,10}\cdot\frac{\vartheta_{i,set}-\vartheta_{i,0}}{\vartheta_{i,10}-\vartheta_{i,0}} \tag{5}$$

with $\Phi_{hc,nd} = \varphi_{hc,nd}\cdot A_{nfa}$

If the indoor air temperature is higher than the set temperature for heating (e.g. $\vartheta_{i,set,h} = 20$ °C) and lower than the set temperature for cooling (e.g. $\vartheta_{i,set,c} = 26$ °C) the heat flux $\Phi_{hc,nd}$ is 0. If the heating and cooling capacity is limited due to the

technical facility, the heating and cooling heat flux might be limited to $\Phi_{h,max}$ or $\Phi_{c,max}$, respectively. Hence, the actual heating or cooling energy $\Phi_{hc}$ is recalculated with the net floor area for one of these five cases:

–    $\Phi_{hc} = 0$ W m$^{-2}$ if $\vartheta_{i,set,h} < \vartheta_i < \vartheta_{i,set,c}$

    –    $\Phi_{hc} = \Phi_{hc,nd}$ if $\Phi_{hc,nd} < \Phi_{h,max}$ in heating mode, or $\Phi_{hc,nd} < \Phi_{c,max}$ in cooling mode, respectively.

    –    $\Phi_{hc} = \Phi_{h,max}$ if $\Phi_{hc,nd} > \Phi_{h,max}$ in heating mode, or $\Phi_{hc} = \Phi_{c,max}$ if $\Phi_{hc,nd} > \Phi_{c,max}$ in cooling mode, respectively.

With the heating or cooling energy $\Phi_{hc}$ [W] the actual temperatures $\vartheta_m$, $\vartheta_s$, and $\vartheta_i$ are calculated from Eq. (1) to (3).

From these simulation results we calculate the operative room temperature, the final energy demand for heating and cooling,
the anthropogenic waste heat and the heat flux from the room to the façade.

The operative room temperature $\vartheta_o$ is the uniform temperature of an imaginary black room in which a person exchanges the same heat through radiation and convection as in the existing non-uniform environment. $\vartheta_o$ is used for the evaluation of the indoor climate and is calculated from the indoor air temperature $\vartheta_i$ and the surface temperature $\vartheta_s$ according to ISO 13790 (2008):

$$\vartheta_o = 0.3 \cdot \vartheta_i + 0.7 \cdot \vartheta_s \qquad (6)$$

The operative room temperature $\vartheta_o$ is calculated as a mean value of the room air temperature and the respective mean radiation temperature of the room surfaces, in which the proportion of convection (indoor air temperature $\vartheta_i$, i.e. 0.3) and radiation (surface temperature $\vartheta_s$, i.e. 0.7) depends mainly on the prevailing air velocity, cf. EN 15251 (2007).

The model according to ISO 13790 (2008) was validated with monitoring data (simulation-measurement validation) and
other simulation programs (cross-model validation). The accuracy of the advanced analytical model has been compared repeatedly with numerical simulation models with special respect to uncertain input parameters, different building technologies, and stochastic user behaviour (Burhenne et al., 2010).

Based on the building energy concepts and the input parameters from the model database, the electrical energy demand (e.g. for lighting, ventilation and office / residential equipment), the heating energy demand (e.g. heat pump systems, boilers,
cogeneration or solar thermal energy) and the cooling energy demand (e.g. compression or thermally driven chillers, adiabatic cooling, cooling towers, ground cooling) are calculated.

The final energy demand for heating and cooling $\Phi_{hc,f}$ is given in electrical and / or fuel energy and depends on the energy efficiency of the technical facility $e_{f,hc}$, e.g. from DIN V 18599 (2011):

$$\Phi_{hc,f} = \Phi_{hc} \cdot e_{f,hc} \qquad (7)$$

The operative room temperature $\vartheta_o$ is a strong indicator for thermal comfort and the final energy demand $\Phi_{hc,f}$ for the efficiency of the building and its technical facilities. These output variables can be used favourably for the analysis of the buildings in urban environments and the evaluation of heat stress in summer, or the decrease of heat energy and the increase of cooling energy due to the urban heat island effect.

Considering the thermal energy demand for heating and cooling, the anthropogenic heat production is calculated and passed
back to PALM (cf. Figure 2). The anthropogenic waste heat $\Phi_{hc,w}$ strongly depends on the energy supply system (e.g. district heating / cooling, heat pump, thermally driven or compression chiller, boiler) and is calculated from a waste heat coefficient

$q_{hc}$ according to DIN V 18599 (2011). $q_{hc}$ is zero for district heating / cooling, positive for boilers or chillers and negative for heat pump systems.

$$\Phi_{hc,w} = \Phi_{hc} \cdot q_{hc} \qquad (8)$$

Figure 2 shows the integration of the building indoor model into the PALM model. The heat transfer through the façade and the surface temperature of the facade is calculated by the building surface model (Resler et al., 2017). Due to the radiative heat transfer through transparent façade elements and the heat storage in opaque façade elements, the interface between the two modules is slightly different for windows and walls:

    –   For windows, the whole energy balance is calculated by the solar heat gain coefficient g [-] and the shading factor $F_C$ for

165          shading devices for heat gains and by the overall heat transfer coefficient U [W m$^{-2}$ K$^{-1}$] for the energy transport due to the temperature difference between inside and outside, since the complex long and short-wave radiation processes within the window and at its surfaces are considered in these characteristics U and g. As the overall heat transfer U is calculated only with the exterior and the room temperature, there is no feedback from the building surface model to the building indoor model. Hence, the heat flux from the inside to the window is

170        $$\Phi_w|_{window} = H_{t,es} \cdot (\vartheta_e - \vartheta_s) \qquad (9)$$

    –   For walls, the building surface model gets the specific heat flux through the exterior walls as an input from the building indoor model for the simulation of façade outside surface temperature.

         $$\Phi_w|_{wall} = H_{t,wm} \cdot (\vartheta_w - \vartheta_m) \qquad (10)$$

         The building indoor model gets the wall temperature $\vartheta_m$ at the last time step t-1 from the building surface model, cf.

175        Figure 3:

         $$\vartheta_{w,t}\big|_{building\ indoor\ model} = \vartheta_{n4,t-1}\big|_{urban\ surface\ model} \qquad (11)$$

The building surface model is based on four wall layers, since almost all traditional or innovative façade construction can be modelled accurately with four layers, e.g. wood panel – wood wool – timber frame construction – plasterboard (lightweight construction) or mortar plaster – thermal insulation composite system – concrete – plaster (heavyweight construction) from

outside to inside. The outside surface temperature is calculated from the energy balance at the façade and the heat flow into the façade is given by the building indoor model. Considering the outside surface temperature and the inside heat flow as driving forces, the wall model is numerically based on a 4R4C-model. Figure 3 shows the with five temperatures (outside surface temperature $\vartheta_{s,o}$ and four wall temperatures $\vartheta_{n,i}$), four serial heat transfer coefficients H [W K$^{-1}$] (as the reciprocal value of R [K W$^{-1}$]) and four heat-storage capacities C [J K$^{-1}$] for each wall layer.

**3 Model database**

A model database is used for the parametrization of the building indoor model and the building surface model. The database provides building physical parameters of the building envelope, geometry data and operational data (incl. user behaviour,

control strategies and technical building services). The only available building information is often the age of the building, its construction material of façade and coating, the façade and window area, and the cubature. Hence, the model database defines all building physical parameters and operational data based on those basic parameters according to a building typology (IWU, 2018). The model database contains four areas:

- The building description is based on geometry, fabric, window fraction and ventilation models.
- The user description is based on (stochastic) user models regarding window opening and use of solar control, and user profiles regarding attendance and internal heat gains.
- The person description is based on the metabolic rate and the clothing value.
- The HVAC energy supply system is simulated with simplified models based on characteristic line models (considering the normative standards with regard to efficiencies) for different heating, cooling, ventilation and air-conditioning concepts. The model database contains also operation strategies for the energy supply system.

The input information on building physical parameters from a regional survey or an urban planning tool is often uncertain and inconsistent. The model database is well-structured and includes sub-models which process information on different levels of accuracy and precision. Hence, the database is built up on a standardized building topology and can manually be adapted in order to evaluate measures with regard to the façade or to the building energy supply.

The standard database contains six building types according to the German building topology (IWU, 2018), i.e. building age from the 1920s, 1970s and the 1990s for residential and non-residential buildings. The summer heat protection corresponds to the minimum requirements with regard to DIN 4108-2 (2013). Typical attendance and internal heat gains are taken from DIN V 18599 (2011) and empirical values (Voss et al., 2006).

## 4 Integration into the PALM model system

The building geometry and the resolution given by the PALM model define both the volume of the building $V_{building}$ and the number of façade elements $n_{facade}$. Each building indoor model contains as many indoor volumes $V_{indoor}$ as façade elements. Thus, all global parameters (i.e. air change per hour, internal heat gain per net floor area and heat capacity) are referred to this virtual room volume:

$$V_{indoor} = \frac{V_{building}}{n_{facade}} \tag{12}$$

While PALM uses an adaptive timestep resolution (which is typically in the order of a few seconds or less), the building indoor model is run for each hour of the day. The results (i.e. indoor environment, surface temperatures, and anthropogenic waste heat from building operation) are fixed for the next hour.

Figure 4 shows simulation results for a summer and a winter simulation run in a typical urban situation with street canyons, block development and high-rise buildings, parks and water. The area around Ernst-Reuter-Platz in Berlin (Germany) contains residential and non-residential buildings from different constructions years (simplified to 1920s, 1970s and 1990s

building standard) with different heating (i.e. district heating or boiler) and cooling facilities (i.e. compression chiller or no mechanical cooling). All parameters for building physical properties, user profiles and technical facilities are taken from the model data base.

Both graphs show the (local) temperature distribution in the building and around the building and the anthropogenic waste heat from heating and cooling at 11 a.m. All simulation results are shown at 10 m above ground and clearly show that the combined simulation of urban atmospheric conditions, energy balance at the wall surface, and the indoor energy balance yield detailed information on indoor and outdoor temperatures, surface temperatures and energy demand (not shown in the graph) and heat flows from the building's energy system to the urban environment:

- The outdoor temperature $\vartheta_e$ is around +24 °C in the summer scenario (above) and -10 °C in the winter scenario (below) and is locally calculated by PALM that predicts the fluid dynamic and thermodynamic effects of the urbanized area around Ernst-Reuter-Platz in Berlin.

- The operative room temperature $\vartheta_o$ is around 26 °C in the summer scenario (in buildings with active cooling) and around 20 °C in winter due to active heating. There is a remarkable temperature range in buildings with no active cooling: in this summer scenario, the operative room temperature in some buildings rise to 33 °C due to high solar and internal heat gains while other buildings stay at 22 °C due to their high thermal inertia and passive cooling strategies.

- The energy demand for heating and cooling $\Phi_{hc}$ of each volume element depends strongly on the temperature difference between inside and outside, the wind speed at the façade, the building construction and window-to-façade ratio, and the solar radiation and the orientation of the building. The (final) energy demand $\Phi_{hc,f}$ considers the building's energy supply system. Based on the energy conversion factors for each heating or cooling system, the anthropogenic waste heat from the building $\Phi_{hc,w}$ is calculated for each façade element separately and is transferred to the urban area via the outside surface. Façade elements with no anthropogenic waste heat (i.e. buildings with district heating in winter or passive cooling in summer, respectively) are shown in black. The specific waste heat from the building $\varphi_{hc,w}$ due to energy losses of the heating supply system ranges between 2 and 7 W m$^{-2}_{facade}$ in winter and due to the recooling systems of the cooling supply system between 20 and 60 W m$^{-2}_{facade}$ in summer.

Comparing the summer and winter scenario in two buildings shows noticeable differences in Figure 4. Building A is a residential building from the 1920s with a boiler for heating and passive cooling in summer. Building B is an office building from the 1970s with district heating in winter and air conditioning in summer.

- In winter at 11 a.m., the indoor temperature in both buildings is 20 °C since both buildings are heated. However, the building B does not transfer any waste heat to the urban environment since it gets district heating. On the other hand, the heat losses from the heating system in building A results in approx. 4 W m$^{-2}_{facade}$ at this time step. Noteworthy, both buildings are heat sources for the urban environment – independent from the waste heat from the heating systems – due to the higher indoor temperatures.

- In summer at 11 a.m., the indoor temperature in building A is still at approx. 24 °C due to passive cooling during night and the high thermal inertia of the 1920s residential building. Building B has higher solar and internal heat gains and is cooled down to 26 °C. As building A is not mechanically cooled, there is no waste heat from any technical devices. On the other hand, the compression chiller in building B generates approx. 25 W $m^{-2}_{facade}$ waste heat.

These selected examples clearly indicates how the building structure, the building operation, and the heating and cooling system contribute to the urban heat island effect in winter and especially in summer.

**5 Summary**

An analytical solution of Fourier's equation is used to simulate the transient energy balance of a building. This building model is separated into virtual control volumes which are geometrically connected with the atmospheric model. Thus, each simulated building consists of as many control volumes as the number of façade elements connected to the exterior environment. Since the energy balance is numerically solved together with the building surface model (i.e. façade temperature and energy balance of convective, long and short wave radiation, transmission, and energy storage) and the atmospheric model (i.e. air temperature of the first node connected to the respective façade element and façade-near air temperature) the coupled energy flow to and from the building to the urban atmospheric conditions can be analysed.

From an application perspective, the indoor environment and the energy need of the building can be calculated as a function of atmospheric environmental conditions. On the one hand, the heat transmission due to the temperature difference between indoor and outdoor environment, the heat transport due to ventilation and the radiative heat transfer at the building surface results in a strong interaction between the built environment and the urban atmospheric conditions. On the other hand, the energy supply of each building results in an anthropogenic waste heat which heats up or cools down the urban microclimate.

First simulation results from a simulation study for the Berlin city centre show the impact of buildings defined by different building physical parameters and with different technical facilities for ventilation, heating and cooling on the urban atmospheric conditions.

Today, an urban microclimate analysis is usually based on monitoring campaigns and simulation runs for the urban atmospheric conditions. Only a few investigations take the buildings explicitly into consideration. Hartz et al. (2020) use four climate datasets based on test reference years and five building types in order to evaluate the outdoor and indoor heat stress in a German city. Though this sophisticated analysis results in a plausible and extensive vulnerability analysis for the different city districts, it does not reveal the dynamic interaction between the built environment and the urban atmospheric conditions. The building indoor model provides a powerful and easy-to-be-used tool for the evaluation of buildings in the urban microclimate and the impact of the building and its operation on the urban atmospheric conditions at the same time.

**Code and data availability.** The PALM model system 6.0, including the building indoor model, can be freely downloaded from https://palm.muk.uni-hannover.de/trac (last access: 23rd March 2021). The distribution is under the terms of the GNU

General Public License (v3). More about the revision control, code management and versioning of the PALM model system 6.0 can be found in Maronga et al. (2015).

**Author contribution.** Jens Pfafferott developed the building indoor model code with support from Sascha Rißmann, Matthias Sühring, Farah Kanai-Sühring, and Björn Maronga. Sascha Rißmann performed the simulation with support from

Farah Kanai-Sühring and Matthias Sühring. Jens Pfafferott prepared the manuscript with contributions from all co-authors. Sascha Rißmann submitted the manuscript.

**Competing interests.** The authors declare that they have no conflict of interest.

**Acknowledgments.** We would like to thank the two anonymous reviewers for their helpful comments on the manuscript.

The German Aerospace Center (DLR) Project Management supported the consortium.

**Financial support.** This study has been supported by the German Federal Ministry of Education and Research (BMBF) under the grants no. 01LP1601A and 01LP1601C. The Urban Climate Under Change [UC]2 programme is funded by the German Federal Ministry of Education and Research under grant 01LP1601 within the framework of Research for Sustainable Development (FONA; https://www.fona.de/en/, last access: 23rd March 2021), which is greatly acknowledged.

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

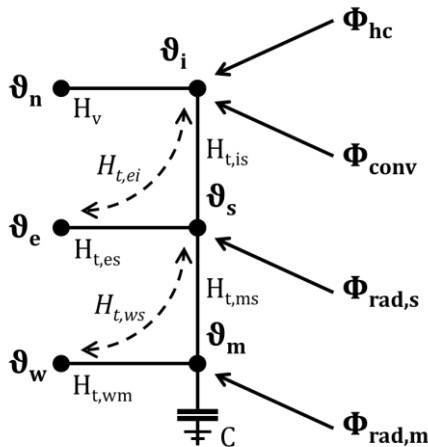

**Figure 1: Building indoor model. Heat flows $\Phi$ and temperatures $\vartheta$ in the R5C1 network. The dashed lines show the energy flow $\Phi_w$ between the building indoor model and the building surface model.**

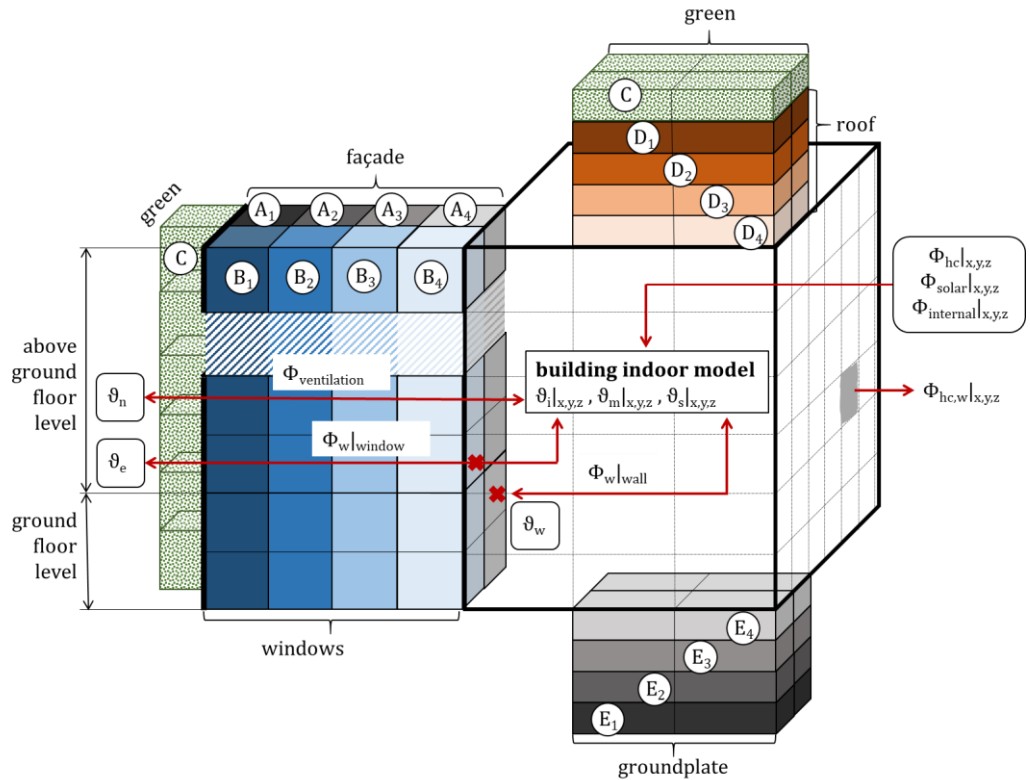


**Figure 2: Integration of the building indoor model (red darts) with the independent driving forces $\Phi_{solar}$, $\Phi_{internal}$, and $\Phi_{hc}$ into the building surface model ($\vartheta_w$) and the PALM model ($\vartheta_n$ and $\vartheta_e$) considering the coupled driving forces $\Phi_{ventilation}$, $\Phi_w|_{window}$, and $\Phi_w|_{wall}$. A- E are five façade types (consisting of different materials) and 1-4 are the four layers for each façade. Additionally, the**

heat transfer due to anthropogenic waste heat (from the cooling and heating facilities) via the façade layer to the PALM model is
shown exemplarily on the right side.

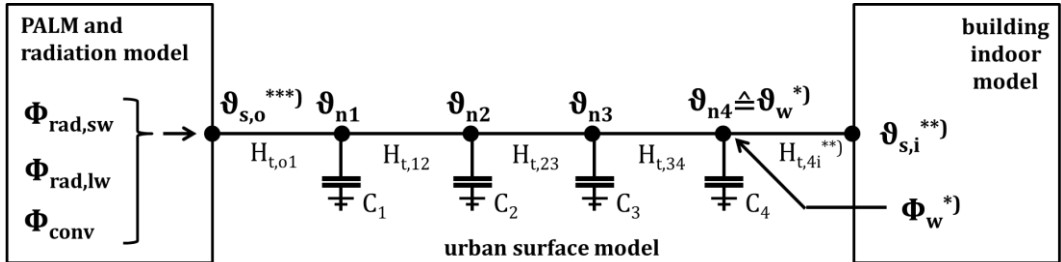

**Figure 3:** Building surface model. Heat flows $\Phi$ and temperatures $\vartheta$ in the R4C4 network. \*) The wall temperature $\vartheta_{n4}$ is the model output to the building indoor model $\vartheta_w$, and the heat flow into the wall $\Phi_w$ is the input from the building indoor model. \*\*) The heat flow at the inside surface is calculated by the building indoor model. Hence, $H_{t,4i}$ is equal to $H_{t,ws}$ in the building indoor model. \*\*\*) The temperature at the outside surface $\vartheta_{s,o}$ is calculated with input variables from the PALM model and the radiation model.

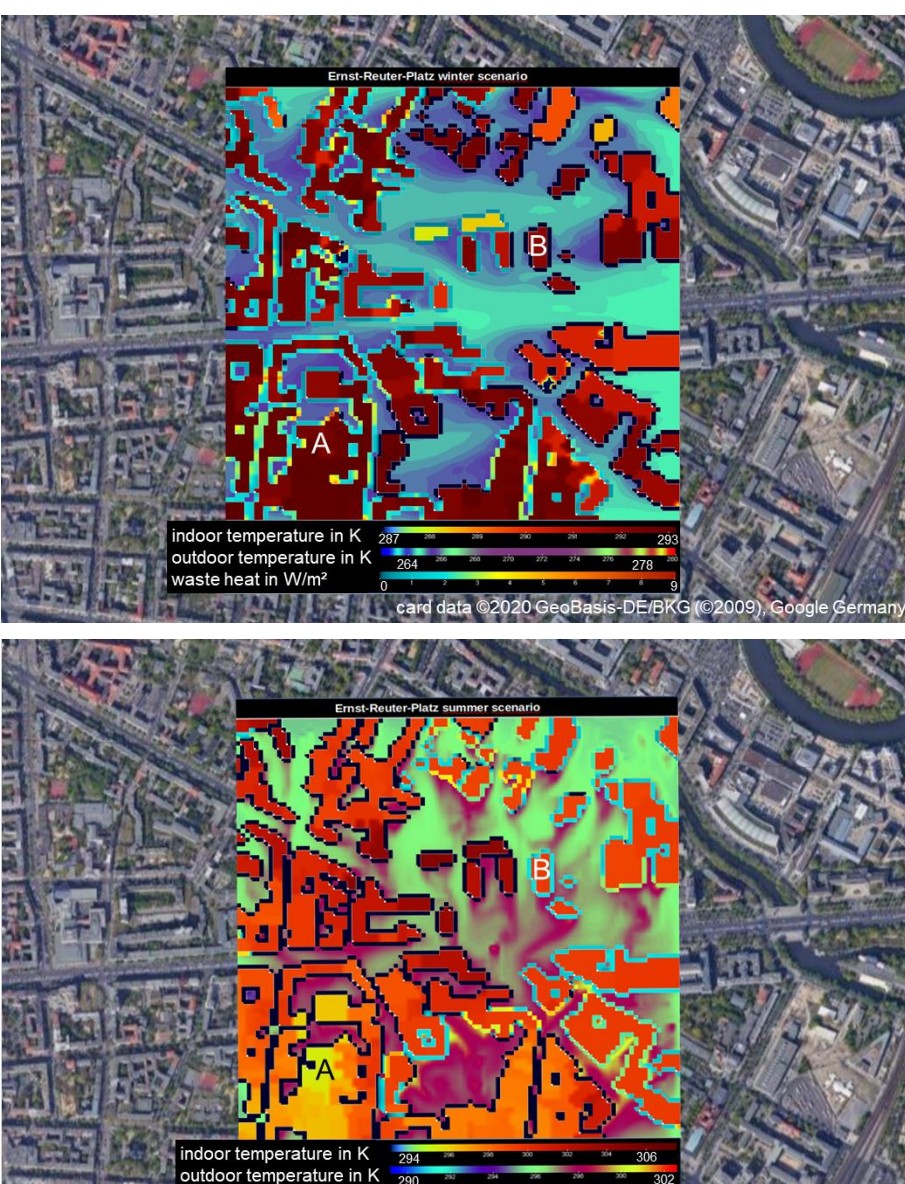

**Figure 4:** **Building surface and building indoor model in a test setup. The Ernst-Reuter-Platz connects five urban street canyons and is surrounded by high-rise buildings. The simulation runs with a resolution of 1 m x 1 m x 1m. The graphs show the operative room temperature $\vartheta_i$, the ambient air temperature $\vartheta_e$ and the anthropogenic waste heat $\Phi_{hc,w}$ at the outside surface. The outdoor temperature is around +24 °C in the summer scenario (above) and -10 °C in the winter scenario (below). Façade elements with no anthropogenic waste heat (i.e. buildings with district heating in winter or passive cooling in summer, respectively) are shown in black.**