# Peer review of "Building indoor model in PALM-4U: Indoor climate, energy demand, and the interaction between buildings and the urban microclimate"

_Geoscientific Model Development, 2020_

## Referee Comment (RC1) · Anonymous Referee #1 · 26 Oct 2020

General comment: The topic of this paper is the implementation of a building model into an urban climate model. The building model representing the transient energy balance of a building is described well; it is based on DIN EN ISO 13790 and uses the analytical solution of Fourier's equation. Less well explained is the coupling between the building model and the urban climate through a so-called urban surface model. Specific comments: It should be explained in more detail how the temperature of the building's boundary layer (which is called 'façade near temperature' by the authors) is calculated as it is influenced by the urban climate and the building itself (surface temperatures). The authors should explain/define tf in figure 2. Why is a 3R3C model used for the façade? Wouldn't a 1R1C model be enough? A schematic drawing of the

building with the addressed facades and surfaces would help to understand the boundaries of the indoor model (interior surfaces and structure?) and the interconnection to the exterior surfaces. How is the anthropogenic waste heat passed back to the urban climate model (page 5, line 133)? In figure 2 it is declared as an input into the urban surface model.

Technical corrections: Headlines of chapter 1 and 2 are identical Page 2, line 56: 'indoor air temperature' Page 2, lines 38 and 60: Replace 'façade near temperature' by something like temperature of the building's boundary layer Page 6, lines 177 and 178: 'top' and 'bottom' instead of 'above' and 'below' Page 7, line 195: 'Fourier's' instead of 'Fouriers'

---

## Author Comment (AC1) · 5 Mar 2021

Dear editor, dear reviewer,

thank you for your comment from Oct. 2020 and the positive response. As we have been waiting for a 2nd review, we haven't answered yet. However, we feel uncertain whether we can expect another comment. We explain the integration of the building indoor model into the PALM program in detail (using a new figure) and show the heat flows / temperatures at the interfaces. Furthermore we explain the urban surface model more in detail although this is not in the focus of this publication. Nevertheless, this makes the whole calculation procedure more understandable. Furthermore, we

considered all technical corrections.

Best regards J. Pfafferott and S. Rißmann
* * *

---

## Referee Comment (RC2) · Anonymous Referee #2 · 6 Mar 2021

**General comment**

The paper numerically investigates the connection between the energy balance and the urban surface model with PALM. It is an important step to include it in a numerical flow solver as the urban heat island effect on the energy balance of a building is larger as the effect of climate change. The derivation of the equations is explained well, however, the presentation of the results in section 4 could be improved (e.g. more detailed explanation of impact factors in the model, where can it be seen in the model? Is there a difference in the model temperature prediction between 1920s, 1970s or 1990s building age (differences in winter (or summer) between different buildings). To

Figure 3 a detailed physical interpretation of the reasons (at least of a few ones) should be given)

**Specific questions**

- Eq. 5: Can you give a motivation for the 0.3 and 0.7 coefficients?

- It is not explained that is meant by third and second wall temperature in line 128.

- There is no reference in the text to Fig. 2.

**Minor comments**

- 1 Introduction

- For completeness you should think about stating Fourier's law.

- line 84, 167: 3600 not 3,600

- line 85: $v_m$ not $v$m (same in line 93 with $v$i, same 104)

- Eq. 1: 3600 s in Equation to result in the right unit for $v_m$

- Fig. 3 colorbar figures cannot be identified

- last sentence of section 4 seems to miss something. Should it be 20 and 60 W/m$^2$ in summer?

---

## Short Comment (SC1) · 12 Mar 2021

**Building indoor model in PALM-4U: Indoor climate, energy demand, and the interaction between buildings and the urban climate**

Jens Pfafferott[1], Sascha Rißmann[1], Björn Maronga[2], Matthias Sühring[2]

[1]Offenburg University of Applied Sciences, Institute of Energy Systems Technology, Offenburg, 77652, Germany
[2]Leibniz University Hannover, Institute of Meteorology and Climatology, Hannover, 30419, Germany

*Correspondence to*: Jens Pfafferott (jens.pfafferott@hs-offenburg.de)

**Abstract.** There is a strong interaction between the urban and the building energy balance. The urban climate affects the heat transfer through exterior walls, the longwave heat transfer between the building surfaces and the surroundings, the shortwave solar heat gains and the heat transport by ventilation. Considering also the internal heat gains and the heat capacity of the building structure, the energy demand for heating and cooling and the indoor thermal environment can be calculated based on the urban climate. According to the building energy concept, the energy demand results in an (anthropogenic) waste heat, this is directly transferred to the urban environment. Furthermore, the indoor temperature is re-coupled via the building envelope to the urban environment and affects indirectly the urban climate with a time shifted and damped temperature fluctuation. We developed and implemented a holistic building model for the combined calculation of indoor climate and energy demand based on an analytic solution of Fourier's equation.

**1 Introduction: Interaction between the building and the urban climate in simulation**

Buildings strongly affect the urban climate. And the urban climate strongly affects the indoor climate and energy demand of buildings. A good review on experimental and numerical studies from the 1960s to today is given by Helbig et al. (2013). Hence, urban climate simulation models should contain a powerful building indoor model in order to evaluate the strong interaction between the building and the urban climate.

In a preliminary simulation study, Jacob and Pfafferott (2012) applied different test reference years (Deutscher Wetterdienst, 2014) on different building concepts and operation strategies. These test reference years consider both the climate change and the urban climate effect. The study clearly revealed that the urban heat island effect has a stronger effect on the building energy balance than the climate change. As expected, the building physical parameters of the building envelope (i.e. heat-transfer coefficients, window area related to façade and floor area, fabric, solar shading) and the user behaviour (i.e. attendance, ventilation, use of shading device) strongly affects energy demand in summer and winter and indoor environment for both residential and office buildings. Results from monitoring campaigns confirm these findings, (Kalz et al., 2014) and (Pfafferott and Becker, 2008).

Favourably, those complex interactions between the built environment and the urban climate can be evaluated based on a sophisticated simulation model (Bueno et al., 2012). Within the MOSAIK project (Maronga et al., 2020), we developed a holistic building model for the coupled calculation of indoor climate (i.e. operative room temperature) and energy demand for heating, cooling, lighting and ventilation. The building indoor model is based on an analytic solution of Fourier's equation and is directly integrated into the PALM code (Knoop et al., 2018).

Furthermore, the building indoor model has an interface with the urban surface model (Resler et al., 2017):

- The temperature of the building's boundary layer is calculated by PALM as an arithmetic mean from the surface temperature of the built environment and the air temperature of the connecting air mode. This 
[revised manuscript text omitted]

---

## Author Response (AR2)

**Response to Anonymous Referee #1 26 Oct 2020**

Thank you for your helpful comments on the manuscript.

We considered all comments. Furthermore, we strongly improved the manuscript based on your comments.

5 **Response to Anonymous Referee #2 6 March 2021**

Thank you for your helpful comments on the manuscript.

We considered all comments. Furthermore, we strongly improved the manuscript based on your comments.

**Relevant changes made with regard to the comments:**

10  1. Abstract has been written more precisely.

2.  Fourier's law and its simplifications is described explicitly.

3. Italic letters for variables, normal letters for parameters in the whole document.

4.  Complement of equations in the algorithm: The whole methodology is described (without detailed calculation procedures).

15  5. Heat transfer through windows and walls is documented exactly. Accordingly, Figure 1 and 3 are updated.

6. Integration of the building model into PALM is shown in new Figure 2.

7. Description of relevant effects which can be found exemplarily in the simulation. Therefore, two buildings A and B are shown in comparison in Figure 4.

8. Update of references.

20

**Relevant changes made with regard of the topical editor:**

1. Technical corrections

2. Inserted credits to Figure 4

3. Note for the production process: We used italic typesetting for all time-dependent variables and normal typesetting
25    for all time-independent parameters.